# Meta-QTL Analysis for Yield Components in Common Bean (*Phaseolus vulgaris* L.)

**DOI:** 10.3390/plants12010117

**Published:** 2022-12-26

**Authors:** Osvin Arriagada, Bárbara Arévalo, Ricardo A. Cabeza, Basilio Carrasco, Andrés R. Schwember

**Affiliations:** 1Departamento de Ciencias Vegetales, Facultad de Agronomía e Ingeniería Forestal, Pontificia Universidad Católica de Chile, Santiago 7820436, Chile; 2Centro de Estudios en Alimentos Procesados, Talca 3460000, Chile; 3Departamento de Producción Agrícola, Facultad de Ciencias Agrarias, Universidad de Talca, Talca 3460000, Chile

**Keywords:** *Phaseolus vulgaris* L., QTL, MQTL, performance, seed weight, days to maturity, days to flowering

## Abstract

Common bean is one of the most important legumes produced and consumed worldwide because it is a highly valuable food for the human diet. However, its production is mainly carried out by small farmers, who obtain average grain yields below the potential yield of the species. In this sense, numerous mapping studies have been conducted to identify quantitative trait loci (QTL) associated with yield components in common bean. Meta-QTL (MQTL) analysis is a useful approach to combine data sets and for creating consensus positions for the QTL detected in independent studies. Consequently, the objective of this study was to perform a MQTL analysis to identify the most reliable and stable genomic regions associated with yield-related traits of common bean. A total of 667 QTL associated with yield-related traits reported in 21 different studies were collected. A total of 42 MQTL associated with yield-related traits were identified, in which the average confidence interval (CI) of the MQTL was 3.41 times lower than the CIs of the original QTL. Most of the MQTL (28) identified in this study contain QTL associated with yield and phenological traits; therefore, these MQTL can be useful in common bean breeding programs. Finally, a total of 18 candidate genes were identified and associated with grain yield within these MQTL, with functions related to ubiquitin ligase complex, response to auxin, and translation elongation factor activity.

## 1. Introduction

Common bean (*Phaseolus vulgaris* L. 2n = 2x = 22) is one of the most consumed grain legumes in the world, especially in Latin America and Africa, where it is a central part of the protein intake of people [1,2]. It is the most important species among the genus *Phaseolus* cultivated for direct human nutrition [3], because its grains are a good source of protein, folic acid, dietary fiber, and carbohydrates and is low in fat [4,5]. It is also a rich source of vitamins and minerals, especially potassium, phosphorus, calcium, iron, and zinc and is low in sodium [1,6]. Due to its nutritional quality and the consumer demand for healthier foods [7], the world production of common beans reached 27.5 million metric tons in 2020, cultivated on a global area of about 34.8 million hectares, where the main producing region was Asia, followed by America and Africa [8].

High grain yield is one of the major objectives in most plant breeding programs globally due to the great interest in increasing the yield potential of crops to meet the growing demand for food as a result of the increasing human population worldwide [9]. Seed yield and their components are complex traits, meaning that many genes with small effects contribute to the phenotypic expression of these traits that are highly influenced by environmental factors [10]. Therefore, understanding their genetic architecture is crucial to establish effective selection criteria and breeding strategies that allow the accumulation and incorporation of loci associated with yield-related traits into the new cultivars. On the other hand, drought is a main constraint for common bean production, and it has been recognized as the main abiotic stressor for its cultivation in rainfed production systems worldwide [11]. Thus, many efforts have been devoted to obtain cultivars tolerant to drought or that are able to escape terminal drought [5,11,12].

The domestication process of common bean led to the generation of two geographically and genetically differentiated wild gene pools: Mesoamerican from Central America, and Andean from South America [13,14,15,16]. These gene pools differ in genetic diversity, seed quality, nutritional traits, resistance to diseases, tolerance to abiotic stress, and morphologic and agronomic traits [17], among others. Consequently, the selection of the most productive cultivars in common bean is a difficult task since there are morphological variations, differences in growth habit, bean seed size, and grain color [18,19]. In addition, the selection made during its domestication generated landraces with specific traits that differ slightly in the genetic composition from the wild gene pools [19]. Even so, the ideotype of a modern common bean crop with desirable agronomic traits corresponds to determinate growth habit, time-synchronized flowering, high number of pods per plant, and high grain weight, thus high harvest index. Furthermore, it is desirable to have a short growing period from sowing to harvest to escape critical periods of terminal drought in order to increase grain yield.

One strategy to accelerate the selection of new common bean cultivars containing superior agronomic traits is using quantitative trait loci (QTL) mapping, which has enabled the identification of genomic regions responsible for the observable variability of some important yield-related traits in common bean [20,21]. In fact, numerous studies have been carried out to identify QTLs for yield components in common bean using biparental and diversity panel in association mapping studies [22,23,24]. For example, a total of 39 yield-related QTL were identified on distinct chromosomes in one study, responsible for 5 to 32% of the phenotypic variance for the measured traits, concluding that QTL with single-locus additives and epistatic effects were involved in the control of bean seed size and yield-related traits, whereas the effects of epistatic interactions were smaller than single-locus QTL [25]. Another work reported a total of six QTL for phenological traits, with the majority being for time to maturity rather than time to flowering [26]. In addition, Valdisser et al. [27] performed a genome-wide association (GWAS) analysis in a Mesoamerican panel of common bean germplasm to identify genomic regions associated with yield under two water regimes. A total of 185 and 18 QTL associated with seed weight were identified of the crop grown under irrigated and drought conditions, respectively. Moreover, several QTL were mined within or near candidate genes playing important roles in yield, giving insight into the genetic mechanisms underlying these traits. However, the individual QTL identified in biparental populations are associated with particular genotypes and environmental conditions; therefore, they are often not useful in genetic improvement programs.

Meta-QTL (MQTL) analysis gathers information from many studies and enhances QTL locations by narrowing down the confidence intervals obtained from individual studies and correlating them with each other [28]. MQTL analysis encompassing previously identified QTLs for specific traits has been carried out in crops such as rice [29], wheat [30], maize [31], and barley [32]. However, there are only few MQTL analyses available in common bean for different traits including white mold resistance [33], seed Fe and Zn concentrations [34], anthracnose resistance [35], and resistance against fungal and bacterial diseases [36]. However, to the best of our knowledge, MQTL of yield-related traits has not been previously accomplished in common bean, conferring high novelty to this manuscript. The main objective of this study was to identify yield-related meta-QTL in common bean with the purpose of introgressing them into new materials using traditional and/or modern breeding techniques. This will allow common bean breeding programs to produce and release new high-yielding genotypes that will in turn benefit the farmers due to higher profitability of the crop.

## 2. Results

### 2.1. QTL for Yield-Related Traits

To identify consensus genomic regions associated with common bean yield-related traits, a total of 21 studies published in the last 16 years (from 2006 to 2021) were considered in this work. These studies comprise 25 different biparental populations with a total of 47 parental lines. The size of mapping populations varied from 75 to 437 lines derived from Andean and Mesoamerican germplasms. Details on these mapping populations are summarized in Table 1. 

A total of 667 QTL distributed throughout all 11 chromosomes (Pv01 to Pv11) were collected for 17 traits evaluated (Table 2 and Appendix A). The number of QTL for each trait and their distribution on 11 common bean chromosomes are shown in Figure 1. In general, the number of QTL per chromosome varied from 26 on chromosome Pv10 to 108 on chromosome Pv01, with an average of approximately 61 QTL per chromosome. Specifically, a total of 208 QTL were found for phenological traits (155 and 53 QTL for days to flowering [DTF] and days to maturity [DTM], respectively), whereas a total of 459 QTL were identified for yield-related traits, with 100-grain weight (100SW; 182 QTL) and yield (Yd; 114 QTL) being the traits with the highest number of QTL identified. Individually, the QTL explained between 0.3 and 64% of the phenotypic variance, with an average of 14%. In addition, a total of 135 QTL had a value greater than 20%; therefore, they were considered as major QTL. Finally, the confidence intervals for each QTL ranged from 0 to 103.16 cM, with an average of 14.87 cM. 

### 2.2. Consensus Genetic Map and Projection of QTLs

The high-density linkage map developed by Song et al. [56] was used as a reference to develop the consensus map. The consensus map contained a total of 7876 markers with a length of 1853.5 cM (Appendix A). The genetic length ranged from 103.1 cM on chromosome Pv10 to 260.3 cM on chromosome Pv02. The number of markers per chromosome varied from 270 on chromosome Pv06 to 1038 on chromosome Pv11. 

Of the collected QTL, 64.76% (432) were projected on the consensus genetic map (Figure 2). A total of 137 and 295 QTLs were projected for phenological and yield traits, respectively. The remaining 235 QTL were not projected because they lacked common markers between the original and the consensus map, and/or the QTL showed low PVE causing a large CI. The chromosome Pv01 had the highest number of projected QTL (73), while the chromosome Pv10 had the lowest number of projected QTL (13).

### 2.3. Meta-QTL Detection

With the 432 QTL projected on the consensus map, a total of 280 QTL were grouped into 42 MQTL, which contained QTL from at least two different studies (Table 3). One hundred fifty-two QTL remained as single QTL since they did not overlap with any MQTL interval, or the QTL overlapped with more than one MQTL due to their large CI. The chromosome Pv08 was the one that grouped the highest number of MQTL (6), while the chromosome Pv11 had the lowest number of MQTL (2). The number of QTL per MQTL varied from 2 on four chromosomes (Pv03, Pv05, Pv08, and Pv10) to 19 on chromosome Pv02 (Yd_MQTL2.1) located at 26.45 cM, with an average of 6.67 QTL per MQTL. Another MQTL on chromosome Pv02 (Yd_MQTL2.4), located at 71.28 cM, had a high number of QTL (15) that also contains the largest number of studies (6). Similarly, two MQTL on chromosome Pv07 “Yd_MQTL7.1” and “Yd_MQTL7.2”, located at 21.28 and 52.59 cM, respectively, clustered 16 QTL per MQTL that were reported by 6 (Yd_MQTL7.1) and 5 (Yd_MQTL7.2) different studies. The number of traits involved per MQTL ranged from 1 in Yd_MQTL11.1 at 12.53 cM (Pv11) and Yd_MQTL8.5 at 109.81 cM (Pv08) to 7 in Yd_MQTL7.2 on chromosome Pv07 at 52.59 cM. Although phenological traits such as DTF and DTM were included in the analysis, no MQTL exclusively grouped these traits. The CI at 95% of each MQTL varied from 0.29 to 21.93 cM, with an average of 4.35 cM, which is significantly lower than the average CI (14.87 cM) considering the original QTL.

### 2.4. Identification of the Candidate Genes on MQTL

The genes present in 32 MQTL identified were further investigated. The MQTL “Yd_MQTL3.1”, located on chromosome Pv03 at 17.12 cM with an interval of 6.64 cM, was the region that contained the highest number of genes, due to the breadth of this genomic region. On the other hand, on the MQTL “Yd_MQTL4.1” located on chromosome Pv04 at 24.53 cM (CI: 2.95 cM), only 3 genes were identified in this region. In relation to candidate genes, a literature review determined the relationship between grain yield and functions associated with ubiquitin ligase complex, response to auxin, and translation elongation factor activity [57,58,59]. A total of 18 candidate genes were identified and associated with grain yield. These were found in 8 of the 42 MQTL identified, which were distributed on all chromosomes except Pv05, Pv06, Pv09, and Pv11 (Table 4).

## 3. Discussion

### 3.1. QTLs for Yield-Related Traits in Common Bean

Common bean is grouped into two geographically isolated and genetically differentiated wild gene pools: Mesoamerican from Central America, and Andean from South America [17]. This history of domestication led to differences in genetic diversity, agronomic, morphological, and phenological traits between these gene pools [37,60,61]. The Mesoamerican gene pool is characterized by either small (<25 g 100 seed weight^−1^) or medium (25–40 g 100 seed weight^−1^) seeds [62], whereas the Andean gene pool is characterized by larger seeds (>40 g 100 seeds weight^−1^) [63]. In contrast, the common beans from the Mesoamerican gene pool are characterized by possessing greater drought tolerance than Andean beans [55]. In fact, most Mesoamerican populations have been utilized to identify drought tolerance-associated QTL for yield-related traits [38,44,45,46]. However, limited progress has been made in the transfer of traits related to drought tolerance from Mesoamerican to large-seeded Andean beans [26,41]. This coincides with our results since most of the studies performed crosses between parents of the same gene pool, and only four studies used crosses between different gene pools (Mesoamerican × Andean). 

Common bean production is mainly carried out by small farmers, and it is grown on marginal soils and with low technological level in certain parts of the world [1]. The average yield of smallholder production systems is below the potential yield of the common beans produced more extensively. In general, the average yield in several countries do not exceed 1000 kg of grain per hectare, while yield potential is significantly higher [1]. Consequently, increased grain production should be an imperative due to the urgent need in increasing the yield potential of crops to meet the increasing demand for food resulting from the expanding human population [9]; this also translates into increased income for farmers.

The main approach to augmenting crop yield is to increase the number and the weight of grains [64]. In this sense, the traits that have been studied to increase seed yield in common bean are pods per plant, seeds per plant, seed weight, and pod harvest index [65]. However, 100-seed weight is the most important component in determining the yield [66] because it is less affected by environmental factors than the other yield components [67]. This is consistent with our results since the trait with the highest number of QTLs was 100-seed weight (100SW) followed by pod harvest index (PHI), which have a positive correlation with the following traits: pods per plant and seed per pod [68,69]. However, a negative correlation between pods per plant and 100-seed weight (100SW) was recently reported in common bean, suggesting that a high number of pods per plant reduces seed weight [70]. 

According to the distribution of QTL related to yield through the common bean genome, the chromosomes with the highest number of QTL were Pv01 (108) and Pv02 (80), whereas chromosomes Pv10 (26) and Pv11 (31) were the ones with the lowest number of QTL. This result differs from what was found in the MQTL analysis for white mold resistance [33], in which from a total of 37 QTL, 11 were located on chromosome Pv07. Additionally, Izquierdo et al. [34] identified a total of 87 QTL associated with seed Fe and Zn concentrations, and most of these QTL were located on the Pv06 chromosome. Recently, out of a total of 88 QTL related to anthracnose resistance, the chromosome Pv04 had the highest number of QTL [35]. In addition, a comprehensive multiple disease resistance MQTL was performed to identify MQTLs associated with resistance against common bacterial blight (CBB), halo blight (HB), white mold (WM), fusarium root rot (FRR), anthracnose (ANT), and angular leaf spot (ALS) diseases. A total of 152 QTL were collected in this study and their distribution was mainly on chromosomes Pv01 and Pv07 [36]. These results demonstrate the complexity of the genetic architecture associated with the different traits of interest studied in common bean. However, the Pv10 and Pv11 chromosomes are consistently those with the lowest number of QTL for the different traits, which can be explained by the smaller genetic size (cM) of these chromosomes compared with the other nine chromosomes [17,56]. The Pv10 chromosome has been considered important in the identification of QTL for resistance to angular leaf spot [71]. In fact, a major QTL, ALS10.1 spanning 5.3 Mb and located at the end of chromosome Pv10, has been identified, which explained between 16 to 22% of the phenotypic variation [72,73,74]. Finally, to the best of our knowledge, this is the first MQTL study that examines yield-related traits and that considers the largest number of QTL (667) to perform a MQTL analysis in common bean. 

### 3.2. MQTL for Yield Components in Common Bean

Despite the thousands of QTLs related to yield identified in different plant species, very few of them have been useful in genetic improvement programs [75] due to their minor effects and the influence of the environment [76]. Therefore, the main objective of MQTL analysis is to identify stable QTL in the plant genomes, which can be useful in breeding programs through marker-assisted selection. Previous studies have detected MQTL for different important traits in common bean; however, none of them have identified MQTL for yield-related traits. For example, Vasconcellos et al. [33] defined nine MQTL for resistance to white mold distributed on 7 chromosomes (Pv01, Pv02, Pv03, Pv05, Pv06, Pv07, and Pv08). Izquierdo et al. [34] identified a total of twelve MQTL for seed Fe and Zn concentrations and content across all chromosomes except on Pv03, Pv05, and Pv10 chromosomes. Shafi et al. [35] identified 11 MQTL for anthracnose resistance on 6 chromosomes. Finally, Rahmanzadeh et al. [36] identified nine MQTL for resistance against multiple diseases distributed on 7 chromosomes. In our study, a total of 42 MQTL for yield-related traits were located on all chromosomes. Therefore, this is the first study reporting MQTL for yield components and it is the study with the highest number of MQTL identified in common bean. 

Drought is the main abiotic stress affecting common bean production; therefore, improving adaptation to dry environments has become an important breeding objective due to the increasing scarcity of available water for crops [77]. In this sense, it is desirable to cultivate a short growing period from sowing to harvest that enables the escape of critical periods of terminal drought and ideally increases grain yield, which is particularly important for common bean grown under rainfed conditions. According to Collins et al. [78], phenological traits are important for drought tolerance studies because of the effect of flowering time on the detection of QTL for yield under drought conditions. In the present study, several QTL associated with phenological traits such as DTF and DTM were considered in the MQTL analysis. In fact, most of the MQTL (28) identified contain QTL associated with yield and phenological traits. This is in agreement with the results obtained by Blair et al. [46], who found that most of the QTL for yield were colocalized with QTL for DTM, and they concluded that these traits would be reliable and effective measures to deal with drought tolerance in common bean. In this sense, these MQTL could be useful in common bean breeding programs, whose objective is to increase yield under drought conditions. However, considering that there is a negative correlation between phenological and yield traits [22,38,41,47,79], these MQTL should be taken with caution. Therefore, a better characterization of the genes present in these MQTL regions and their subsequent application is necessary to evaluate their real usefulness in common bean breeding programs.

Most of the Andean cultivars are sensitive to photoperiod [47] and generally, the lines have not been able to combine their short phenological cycle with higher seed yield [79,80,81]. In another study, photoperiod-sensitive genotypes flowered and matured later; therefore, they had an extended vegetative growth stage and accumulated more biomass than the photoperiod-insensitive genotypes [22]. In addition, 14 MQTL were exclusively identified for traits related to yield components, so they can be used in breeding programs where the goal is to increase yields regardless of the environment used.

### 3.3. Candidate Genes Underlying the MQTL

Grain yield is a complex trait because it is affected by the environment and the action of multiple genes with minor effects. The dissection of genome regions related to yield-related traits has been tackled through linkage analysis between molecular markers and phenotypic variants. This has allowed the identification of many QTL related to the production and size of the grains in common bean [55,82,83,84]. For example, Peréz-Vega et al. [83] identified five QTL that together explained 74% of the seed length variation. MQTL analysis is a genetic technique that allows the integration of the information available from linkage analysis and identification of QTL to refine the QTL locations by narrowing the confidence intervals [85]. The results allowed the identification of 18 candidate genes related to grain yield in 8 MQTL distributed on 7 out of the 11 chromosomes of common bean, in which the largest number of genes were positioned on chromosomes Pv07 and Pv10 (Appendix A). From them, 12 candidate genes were homologous to *AUXIN RESPONSE FACTOR 2* genes identified in *Arabidopsis thaliana*, which have been related to an extra production of cells in the seed coat and would substantially increase flower set and flower formation of pods [57]. In addition to playing a role in the process of plant growth and development, auxin response factors (ARFs) enhance the tolerance to salt and drought in transgenic *Arabidopsis* [86,87].

Similarly, five genes homologous to the tobacco elongator-associated protein were identified. These genes, co-expressed with the gene that encodes for RD21-like protease, would be related to the production of seeds and biomass [58]. In addition, it was possible to find the Ring-type E3 ubiquitin ligase homologous gene, which has been related to an alteration in the shoot system with an increasing number of side shoots and seed production in barley [59]. According to the transcriptomic results available in Phythozome for common bean, the auxin response genes are expressed at a low level in all tissues. In contrast, the elongator-associated protein and ubiquitin genes are mostly expressed in mature green pods. Consequently, the 8 MQTL are the most valuable genome regions for common bean breeding because they contain genes that can potentially affect and increase grain yield.

Finally, the MQTL identified in this study contain candidate genes involved in the ubiquitin ligase complex, response to auxin, and translation elongation factor activity process. In the future, further characterization of these genes and their subsequent evaluation on the effects on the phenotype of interest is highly necessary for a better understanding of the utility of these MQTL detected in common bean.

## 4. Materials and Methods

### 4.1. Bibliographic Review and QTL Collection

An exhaustive bibliographic review was carried out to find studies reporting QTLs associated with yield-related traits in common bean in the last 16 years. Keywords such as “bean”, “*Phaseolus*”, “QTL”, and “yield” were used for the searches in the databases of Google Scholar and PubMed. A total of 17 traits related to yield component were considered in this study (Table 2). Two phenological traits were considered: days to flowering (DTF) and days to maturity (DTM). Additionally, 15 yield-related traits were considered: plant height (PH), plant width (PW), yield (Yd), pods per plant (PPP), seeds per plant (SPP), 100-seed weight (100SW), pod length (PL), pod number per area (PNA), seed number per area (SNA), number seed per pod (SPPd), pod harvest index (PHI), pod weight (PdW), seed width (SWi), seed length (SL), and seed yield per plant (Syd). For the MQTL analysis, only studies that presented the type and size of the mapping population, chromosome name, positions of QTL (peak position and/or confidence intervals), LOD (logarithm of the odds) score or LRS (Likelihood ratio statistic) values for each QTL, and percentage of phenotypic variance explained for each QTL (PVE or *r*^2^) were considered. Each QTL was treated as an independent QTL, even if some of them were detected in multiple environments or genetic backgrounds. When the QTL position peak was not reported, it was calculated as the average of the CI. Conversely, if the CI was not reported, it was calculated according to Guo et al. [88]. LRS scores were converted into LOD values by a direct transformation where LRS=4.6× LOD score [89]. 

### 4.2. Consensus Genetic Map and Projection of QTLs

Currently, the linkage map developed by Song et al. [56] has the highest density of molecular markers in common bean and it was used as a reference map. This high-density genetic map was constructed using a large F_2_ population derived from the cross between Stampede × Red Hawk varieties. The map consisted of 7040 markers distributed across the 11 chromosomes (Pv01–Pv11), spanning 1042 cM, and a density marker of 6.75 markers per cm. The software BioMercator V4.2 [90] was used to develop a consensus map by integrating this reference map with the markers linked to QTLs identified from different mapping populations considered in this study, and for subsequent MQTL analysis. 

To project the individual QTL identified in different studies on the consensus map, the original QTL data (QTL position, PVE, LOD score, and CI) was considered. The QTL projection was performed using the QTLProj command available in Biomercator v4.2 following the homothetic approach described by Chardon et al. [91].

### 4.3. Meta-QTL Analysis

For the MQTL analysis, files containing the map and the QTLs’ information for each of the studies considered were prepared according to the requirements of the BioMercator v4.2 software. The MQTL analysis was conducted with the projected QTLs on the consensus map following the approach of Veyrieras et al. [92]. The best model of MQTLs was chosen according to Akaike Information Criterion (AIC), corrected Akaike Information Criterion (AICc and AIC3), Bayesian Information Criterion (BIC), and Average Weight of Evidence (AWE) criteria. The number of MQTLs present on each chromosome was determined when values of the model selection criteria were the lowest in at least three of the five models [64]. Given that MQTL analysis is a method to combine QTL results from different independent analyses, a MQTL with a confidence interval of 95% was reported when it contained QTL from at least two different studies [28].

### 4.4. Identification of Candidate Genes

The candidate genes associated with the MQTL regions were determined in relation to the physical position, which was identified through the left and right markers. The last version of the *P. vulgaris* reference genome v2.1 available in Phytozome (https://phytozome-next.jgi.doe.gov/ (accessed on 20 October 2022)) was used to identify genes into the physical position of MQTL. Genes annotated by Gene Ontology (GO) were selected (Appendix A). To select the candidate genes, a literature search was carried out for the identification of genes associated with grain yield. The main focus was on ubiquitin ligase complex, response to auxin, and translation elongation factor activity. 

## 5. Conclusions

Meta-QTL studies provide valuable information for QTL fine mapping and key genes for cloning. In this sense, we performed a meta-analysis study for yield-related traits in common bean, in which several genomic regions consensus (MQTL) associated with yield-related traits were identified. Among them, the “*Yd_MQTL2.4*”, “*Yd_MQTL7.1*”, and “*Yd_MQTL7.2*” had the highest number of QTL and studies reported, so they can be seen as promising elements for future research and potential use in common bean breeding programs. In addition, some MQTL contained QTL identified for yield and phenological traits, which can be useful to increase yield in common bean breeding programs under drought conditions. Moreover, a total of 18 candidate genes participating in ubiquitin ligase complex, response to auxin, and the translation elongation factor activity process were identified within those MQTL in accordance with the complex nature of the yield. Finally, our results are highly novel and of interest to breeders since it is the first and the most complete study identifying MQTL associated with grain yield in common bean.

## Figures and Tables

**Figure 1 plants-12-00117-f001:**
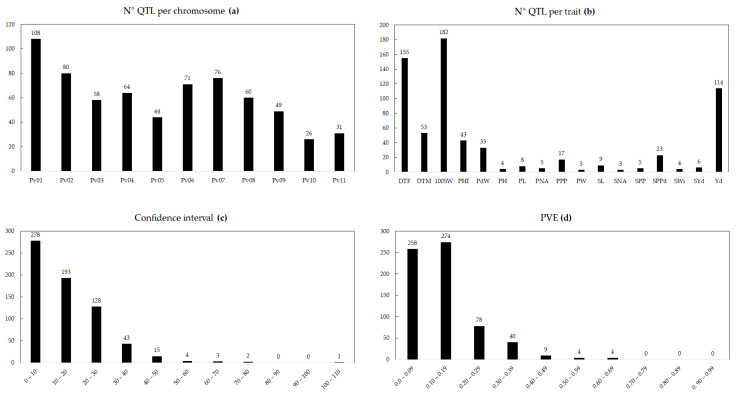
Summary of the 667 collected QTL related to phenological, yield, and yield component in common bean. Number of QTL per chromosome (**a**), number of QTL per trait (**b**), confidence interval (**c**), phenotypic variance explained (PVE) for each QTL (**d**). DTF: Days to flowering; DTM: Days to maturity; PH: Plant height; PW: Plant width; Yd: Yield; PPP: Pods per plant; SPP: Seeds per plant; 100SW: 100-seed weight; PL: Pod length; PNA: Pod number per area; SNA: Seed number per area; SPPd: Number seed per pod; PHI: Pod harvest index; PdW: Pod weight; SWi: Seed width; SL: Seed length; SYd: Seed yield per plant.

**Figure 2 plants-12-00117-f002:**
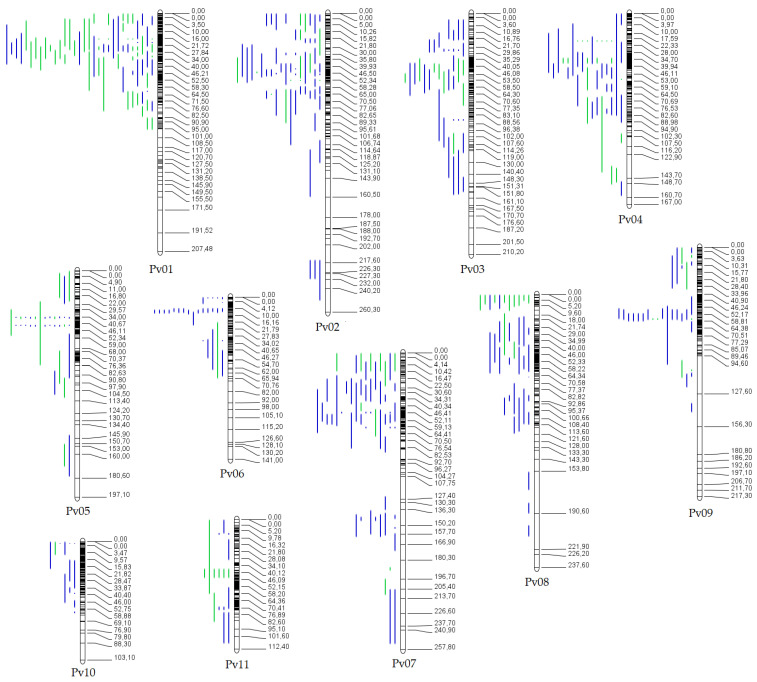
Distribution of the QTL projected throughout all 11 chromosomes (Pv01 to Pv11) on the consensus genetic map. Bars on the left side of the chromosome correspond to QTL related to phenological (green bars) and yield-related (blue bars) traits. Black bars within chromosomes represent marker density. The numbers on the right side of the chromosome correspond to the genetic distance in cM.

**Table 1 plants-12-00117-t001:** Summary of the QTL studies reviewed, including the reference, parental cross, gene pool, type of population, size, traits, number of QTL collected, and the number of environments evaluated in each work.

Reference	Cross	Gene Pool ^a^	Type	Size	Traits	N° QTL	Env
Blair et al. [37]	ICA Cerinza × G24404	A	BC_2_F_3:5_	157	DTF, DTM, PH, PW, Yd, PPP, SPP, 100SW	49	3
Diaz et al. [38]	BAT 881 × G 21212	M	RIL	94	Yd, 100SW, PNA, HI, DTF	53	15
Geravandi et al. [25]	Goli × AND1007	M × A	RIL	100	100SW, SL, SWi, SPPd, PPP, SPP, PL, HI, Syd	26	3
Checa and Bleir [26]	G2333 × G19839	M × A	RIL	84	DTF, DTM, PL, PPP, Yd, 100SW	26	3
Diaz et al. [39]	DOR 364 × BAT 477	M	RIL	98	Yd, PNA, SNA, 100SW, PHI, DTF	25	2
Hoyos-Villegas et al. [40]	3 RIL populations ^1^	M	RIL	76; 48; 36	GY, 100SW, DTF, DTM	14	2
Mukeshimana et al. [41]	SEA5 × CAL96	M × A	RIL	125	Yd, PPP, 100SW, PHI, DTF, DTM	33	3
Wright and Kelly [42]	Jaguar × 115M	N.I.	RIL	96	Yd	3	3
Cichy et al. [43]	Black Magic × Shiny Crow	N.I.	RIL	100	100SW, Yd	8	2
Diaz et al. [44]	Eight elite lines ^2^	M	MAGIC	437	100SW, DTM, DTF, Yd	50	2
Berny Mier y Teran et al. [45]	ICA Bunsi × SXB405	M	RIL	226	DTM, 100SW, Yd, PdW, PHI, SPPd	155	5
Blair et al. [46]	DOR 364 × BAT 477	M	RIL	113	Yd, 100SW, DTM, DTF	51	6
Gonzalez et al. [47]	Bolita × PHA1037	A	RIL	185	DTF	19	12
Sandhu et al. [48]	BK004-001 × H68-4	N.I.	RIL	85	100SW, Yd	19	6
Bassett et al. [49]	Ervilha × PI527538	A	RIL	242	100SW, Yd	17	2
Langat et al. [50]	KAT B1 × GLP2	A	F_2_	120	100SW, Yd, DTM, DTF, PPP, SPPd	16	2
Nabateregga et al. [51]	BRB 191 × SEQ 1027	A	RIL	128	100SW, Yd, HI	7	2
Mkwaila et al. [52]	Tacana × PI 313850	M × A	BC_1_F_4:5_	75	Yd, 100SW, DTF	13	3
Trapp et al. [53]	Buster × Roza	M	RIL	140	Yd, DTM, DTF, 100SW	41	9
Yuste-Lisbona et al. [54]	PMB0225 × PHA1037	A	RIL	185	SWi, SL, 100SW, SPPd	15	6
Sedlar et al. [55]	Tiber × Starozagorski	A	RIL	82	DTF, DTM, SYd, PPP, SPPd, PHI	27	6

DTF: Days to flowering; DTM: Days to maturity; PH: Plant height; PW: Plant width; Yd: Yield; PPP: Pods per plant; SPP: Seeds per plant; 100SW: 100-seed weight; PL: Pod length; PNA: Pod number per area; SNA: Seed number per area; SPPd: Number seed per pod; PHI: Pod harvest index; PdW: Pod weight; SWi: Seed width; SL: Seed length; SYd: Seed yield per plant. ^a^ Mesoamerican (M) or Andean (A) gene pool. ^1^ Three RIL populations: SER48 × Merlot, SER94 × Merlot, and SER95 × Merlot. ^2^ Eight elite founders: SXB412, INB827, ALB213, SEN56, SCR2, MIB778, SCR9 and INB841. Env: number of environments evaluated in the study.

**Table 2 plants-12-00117-t002:** Traits reported in the QTL studies considered in this study.

Category	Trait	Abbreviation
Phenology	Days to flowering	DTF
	Days to maturity	DTM
Yield	Plant height	PH
	Plant width	PW
	Yield	Yd
	Pods per plant	PPP
	Seeds per plant	SPP
	100-seed weight	100SW
	Pod length	PL
	Pod number per area	PNA
	Seed number per area	SNA
	Number seed per pod	SPPd
	Pod harvest index	PHI
	Pod weight	PdW
	Seed width	SWi
	Seed length	SL
	Seed yield per plant	SYd

**Table 3 plants-12-00117-t003:** Details of 42 Meta-QTL (MQTL) identified for phenological, yield, and yield component in common bean.

Chr	MQTL	Peak (cM)	CI (95%)	N QTL	N Studies	Traits	Near Left Marker	Near Right Marker	Physical Interval (bp)
Pv01	Yd_MQTL1.1	5.26	3.19	10	2	DTF, 100SW, PHI, DTM	ss715649523	ss1399949780	3,862,571–2,396,818
	Yd_MQTL1.2	50.74	4.43	5	3	PdW, 100SW, Yd, DTF	ss715648272	ss715646315	47,490,920–48,116,784
	Yd_MQTL1.3	68.28	1.28	4	2	SYd, 100SW, DTF	g1176	IAC259	
	Yd_MQTL1.4	87.09	1.54	5	2	DTF, YSd	ss715645299	IAC76	51,353,193–51,617,802
Pv02	Yd_MQTL2.1	26.45	1.5	19	4	Yd, DTM, 100SW, DTF	ss715647233	ss715639495	2,124,698–2,201,533
	Yd_MQTL2.2	52.93	0.88	4	2	SWi, PdW, DTF	ss715642549	ss1399947721	16,565,668–24,596,150
	Yd_MQTL2.3	58.97	0.58	4	2	SL, DTF	ss715639420	ss715639430	27,998,989–28,312,855
	Yd_MQTL2.4	71.28	0.95	15	6	100SW, PdW, DTM, PHI, DTF, Yd	ss715648503	ss1399947778	27,998,989–36,426,631
	Yd_MQTL2.5	228.21	9.01	3	2	SPPd, 100SW	PVBR94	BMd18	
Pv03	Yd_MQTL3.1	17.12	6.64	5	2	PdW, 100SW, PPP	ss715645941	ss715646941	42,982,727–2,620,445
	Yd_MQTL3.2	48.69	5.68	3	2	Yd, DTM, DTF	ss715640761	ss715647432	21,279,773–32,787,282
	Yd_MQTL3.3	58.75	5.51	6	2	100SW, DTF, DTM	ss715649363	ss715646624	35,509,497–39,662,131
	Yd_MQTL3.4	112.52	12.44	2	2	Yd, DTF	IAC20	ss715646087	51,734,389–52,218,635
	Yd_MQTL3.5	148.83	8.97	4	3	Yd, 100SW	BM197	BM181	
Pv04	Yd_MQTL4.1	24.53	2.95	8	2	100SW, DTF	ss715647356	ss715642594	45,349,844–45,414,255
	Yd_MQTL4.2	42.3	4.51	9	3	Yd, DTF, PHI	ss715648465	ss715644473	7,060,335–35,400,696
	Yd_MQTL4.3	57.71	0.97	8	4	SPPd, DTF, Yd, PNA	ss715648140	ss715647284	41,929,814–3,170,316
	Yd_MQTL4.4	99.76	1.57	10	4	PHI, Yd, DTF, PNA, PdW, PPP	PVBR182	ATA02	45,623,037–45,793,192
Pv05	Yd_MQTL5.1	48.04	0.39	11	2	PdW, DTF, 100SW	ss715644298	ss715641670	9,686,007–18,941,545
	Yd_MQTL5.2	99.76	12.65	3	3	PHI, DTF, DTM	BMd2	IAC286	
	Yd_MQTL5.3	161.68	21.43	2	2	PHI, DTF	Pvat006	IAC261	
Pv06	Yd_MQTL6.1	1.13	0.56	5	3	SPPd, PHI, 100SW	ss715643299	ss715641010	11,458,294–12,875,827
	Yd_MQTL6.2	12.31	1.16	15	2	100SW, PdW, PHI, Yd	ss715639609	E31M50-101	19,087,448–19,325,473
	Yd_MQTL6.3	49.31	10.52	4	2	DTM, 100SW, PPP	ss1399948688	ss715645204	29,060,450–29,310,495
Pv07	Yd_MQTL7.1	21.28	4.94	16	6	100SW, DTF, Yd, PW, PL, SL, SPP	ss715648396	Bng199	220,519–77,369
	Yd_MQTL7.2	52.59	0.95	16	5	100SW, PHI, PdW, SWi, Yd, DTM, PH	ss715640072	ss715642088	10,722,930–34,487,539
	Yd_MQTL7.3	149.2	1.55	10	2	100SW, Yd	PvM40	IAC257	
	Yd_MQTL7.4	187.58	0.83	6	4	SPP, DTF, DTM	IAC232	IAC130	
Pv08	Yd_MQTL8.1	2.6	2.91	11	4	DTF, DTM, PL, Yd	SAS13	sc0002_sc00020141038212_936100	
	Yd_MQTL8.2	32.84	5.76	6	2	100SW, DTM	ss715645836	PVtaaaa1	2,922,586–3,989,686
	Yd_MQTL8.3	56.92	5.24	6	2	DTM, Yd, 100SW	ss715641366	ss1399948977	9,625,469–42,709,886
	Yd_MQTL8.4	94.05	4.08	4	3	100SW, Yd	ss715646536	ss715646503	56,911,339–57,279,814
	Yd_MQTL8.5	109.81	9.28	2	2	Yd	ss715646762	ss715650658	58,783,185–59,613,870
	Yd_MQTL8.6	189.3	2.9	3	2	PPP, 100SW	PvM11	IAC74	
Pv09	Yd_MQTL9.1	12.2	1.93	8	5	100SW, DTM, PPP, SWi	ss715644492	ss715639276	10,826,222–11,082,007
	Yd_MQTL9.2	62.89	0.84	4	2	100SW, DTF	ss715646278	ss715646277	30,662,512–31,043,995
	Yd_MQTL9.3	119.39	1.01	3	2	Yd, 100SW	ss715639364	BM141	37,399,568
Pv10	Yd_MQTL10.1	1.85	0.29	6	3	100SW, DTF, PHI	ss715639843	ss715648665	1,711,719–2,538,255
	Yd_MQTL10.2	27.13	15.76	2	2	100SW, Yd	ss715639433	ss715645511	38,179,211–41,000,138
	Yd_MQTL10.3	45.14	2.65	3	3	Yd, SL	BMc234	Vpe-3	764,184
Pv11	Yd_MQTL11.1	12.53	1.95	4	2	100SW	ss715645472	ss715645474	1,215,049–1,466,213
	Yd_MQTL11.2	76.37	0.54	6	2	PdW, 100SW, DTF, PPP	ss715639694	ss715640406	48,876,825–49,242,323

DTF: Days to flowering; DTM: Days to maturity; PH: Plant height; PW: Plant width; Yd: Yield; PPP: Pods per plant; SPP: Seeds per plant; 100SW: 100-seed weight; PL: Pod length; PNA: Pod number per area; SNA: Seed number per area; SPPd: Number seed per pod; PHI: Pod harvest index; PdW: Pod weight; SWi: Seed width; SL: Seed length; SYd: Seed yield per plant.

**Table 4 plants-12-00117-t004:** List of candidate genes identified and associated with grain yield within the physical intervals of MQTL.

Chr	MQTL	Gene Name	Start (bp)	End (bp)	Description
Chr 01	MQTL1.4	Phvul.001G269400	51,392,676	51,380,889	Translation elongation factor activity
Chr 02	MQTL2.2	Phvul.002G109600	23,522,544	23,521,662	Response to auxin
Chr 03	MQTL3.1	Phvul.003G127801	31,948,232	31,946,345	Response to auxin
		Phvul.003G075200	11,873,729	11,870,757	Translation elongation factor activity
		Phvul.003G127650	31,766,322	31,765,957	Response to auxin
Chr 04	MQTL4.2	Phvul.004G060000	8,421,965	8,419,323	Translation elongation factor activity
		Phvul.004G075100	12,941,701	12,938,447	Translation elongation factor activity
	MQTL4.3	Phvul.004G033100	3,918,987	3,909,164	Ubiquitin ligase complex
Chr 07	MQTL7.2	Phvul.007G219300	34,259,201	34,259,725	Response to auxin
		Phvul.007G219400	34,262,950	34,263,624	Response to auxin
		Phvul.007G219500	34,278,590	34,279,138	Response to auxin
		Phvul.007G219600	34,294,343	34,295,137	Response to auxin
		Phvul.007G219700	34,297,624	34,297,914	Response to auxin
Chr 08	MQTL8.3	Phvul.008G157300	28,399,300	28,376,710	Translation elongation factor activity
Chr 10	MQTL10.2	Phvul.010G117400	39,627,884	39,628,352	Response to auxin
		Phvul.010G117500	39,650,664	39,651,593	Response to auxin
		Phvul.010G125400	40,651,691	40,652,726	Response to auxin
		Phvul.010G127000	40,769,674	40,769,979	Response to auxin

## Data Availability

Not applicable.

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
