# Peer review of "Meta-QTL Analysis for Yield Components in Common Bean (Phaseolus vulgaris L.)"

_plants, 2022, doi:10.3390/plants12010117_

Round 1

Reviewer 1 Report

Please see the file attached.

Author Response

Responses to reviewer 1

Thank you very much for the comments and suggestions of Reviewer 1. We discussed each point thoroughly with the other authors of the paper, and consider that we improved the quality of the paper significantly as a consequence of your input. We addressed most of the reviewers’ suggestions, with some exceptions properly justified. Please see below the answer highlighted in yellow of each one of your questions/remarks.

Abstract

  1. Line 25: Why drought conditions? You have mentioned phenological traits and then you jumped to drought without connection.

R: It was corrected.

  1. Line 28: Key Words can not be the same of the title. Please substitute Commo bean for Phaseolus vulgaris L.; Yield for Performance.

R: It was done.

  1. Line 55: The reference 13 is not the best citation for gene pool characterization. Please consider: Singh et al., 1991; Carović-Stanko et al. 2017; Valdisser et al. 2017; Valentini et al. 2018.

Carović-Stanko, K., Liber, Z., Vidak, M., Barešić, A., Grdiša, M., Lazarević, B., & Šatović, Z. (2017). Genetic Diversity of Croatian Common Bean Landraces. Frontiers in Plant Science, 8, 604. https://doi.org/10.3389/fpls.2017.00604 .

Singh, S. P., Gepts, P., & Debouck, D. G. (1991). Races of common bean (Phaseolus vulgaris, Fabaceae). Economic Botany, 45(3), 379–396. https://doi.org/10.1007/BF02887079

Valdisser, P.A.M.R., Pereira, W. J., Almeida Filho, J. E., Müller, B. S. F., Coelho, G.R. C., de Menezes, I. P. P., Vianna, J. P. G., Zucchi, M. I., Lanna, A. C., Coelho, A. S. G., de Oliveira, J. P., Moraes, A. da C., Brondani, C., & Vianello, R. P. (2017). Indepth genome characterization of a Brazilian common bean core collection using DArTseq high-density SNP genotyping. BMC Genomics, 18(1). https://doi.org/10.1186/s12864- 017-3805-4

Valentini, G., Gonçalves-Vidigal, M. C., Elias, J. C. F., Moiana, L. D., & Mindo, N. N. A. (2018). Population Structure and Genetic Diversity of Common Bean Accessions from Brazil. Plant Molecular Biology Reporter, 36(5–6), 897–906. https://doi.org/10.1007/s11105-018-1129-4.

R: It was modified and the references were included into the revised version of the manuscript.

  1. Line 59: Please change bean size for bean seed size.

R: It was done.

  1. Line 71: The expression “association population” is wrong. You should say Biparental (line 70) and associative mapping (line 71). The plant material used in associative mapping is named diversity panel. We do not refer as population.

R: It was corrected.

  1. Lines 83-84: The statement “However, these individual QTL reported in the literature are associated with particular genotypes…” is not true for GWAS. The diversity panel usually is large enough and contains several cultivars and varieties that are useful for breeding programs. Please re-write the sentence. This affirmative is only true for biparental populations.

R: It was corrected.

Material and Methods

  1. Line 344: We are in 2022 and there are several other studies with higher amounts of SNPs (Beadchip of 12K for common bean is available). Why did the authors restrict so much the reference map to Song et al. (2015) F2 population?

R: The map developed by Song et al. (2015) is one of the densest genetic maps developed to date, and it has been successfully used in various studies. This map was developed using various varieties of common beans; therefore, the markers present in the eleven linkage groups are highly informative in different genetic backgrounds. In addition, this consensus genetic map considers SSR markers that have been widely used to detect QTL in common bean. Considering that MQTL analysis is an approach to combine data sets and for creating consensus positions for the QTL identified in independent studies, this genetic map is useful for this purpose.

  1. Please detail if the QTLs from different studies were classifies into one cluster if the estimated 95% CI had a region in common as Guo et al. (2006). How much you followed Guo´s approach is important to be described.

R: The cluster of QTLs from different studies (or MQTL) were classified according to Veyrieras et al. (2007). The Guo´s approach was used only to determine the confidence interval when this was not reported in the respective study. This is explained in the manuscript in the sections 4.1 and 4.3.

Results:

  1. Lines 141-145: This description should be on material and methods, not in results.

R: The consensus map (Table S2) was elaborated in this study; consequently, it is a result. The BioMercator software was used to develop this consensus map, which integrates the genetic maps from different studies into the reference map developed by Song et al. 2015.

  1. Table 1 and Table 3: the Traits names should be detailed in the baseboard of the table as tables and figures should always be self-explanatory. In the format presented the reader must search in the text for explanation of each trait.

R: The traits names were actually included in the tables.

  1. Line 162: The strategy of considering at least two different studies for grouping the QTls should have been described in Material and Methods. It is not result. The authors should give a reason citing the literature why considering two study model.

R: A sentence and quote was added.

  1. Line 177: It is not clear in the text which is the threshold value for a CI to be considered significative. This should be detailed in methodology.

R: The 95% confidence interval was obtained for each MQTL, and it was added in both the Methodology and the Results sections.

  1. Lines 234 and Line 246-248: Pv11 is the smallest chromosome in bean genome. However, Pv10 has consider an important chromosome for angular leaf spot resistance. Please re-write and consider this fact of Pv10 in your discussion.
  2. a) OBLESSUC, P.R.; MATIOLLI, C.C.; CHIORATO, A.F.; CAMARGO, L.E.A.; BENCHIMOL-REIS, L.L.; MAELI MELOTTO. Common bean reaction to angular leaf spot comprises transcriptional modulation of genes in the ALS10.1QTL. Frontiers in Plant Science, Mar 12;6:152. doi: 10.3389/fpls.2015.00152. eCollection 2015.
  3. b) OBLESSUC, PR; PERSEGUINI, JMC; BARONI, RM; CHIORATTO, AF; CARBONELL, SAM; MONDEGO, JMC; VIDAL, RO; CAMARGO, LEA; BENCHIMOL-REIS, L.L. Increasing the density of markers around a major QTL controlling resistance to angular leaf spot in common bean. Theoretical and Applied Genetics, 126 (10): 2451-2465, 2013.
  4. c) OBLESSUC, PR; BARONI, R; GARCIA, AAF; CHIORATTO, AF; CARBONELL, SAM; CAMARGO, LEA; BENCHIMOL, LL (2012). Mapping of angular leaf spot resistance QTL in common bean (Phaseolus vulgaris L.) under different environments. BMC Genetics, 13:50; http://link.springer.com/content/pdf/10.1186%2F1471-2156-13-50.pdf.

R: A paragraph was added, and the references were included.

  1. Line 244: Reference 32 has a different approach in which concerns the studies considered (The mapping populations ranged in size from 52 to 907 progenies of various types including two backcross (BC), seven F2, and 35 recombinant inbred lines, RIL). They were much more inclusive, and it is difficult to compare results with different methodologies.

R: The work performed in the reference 32 is a MQTL analysis for resistance against multiple diseases, including angular leaf spot (ALS). The objective of the MQTL analysis is to gather information from the QTLs detected in different studies to create consensus genomic regions that are associated with one or several traits of interest, independent of the mapping populations, population size, and genetic background. However, as our focus was to perform a MQTL analysis for yield-related traits, this could explain why the chromosomes with the highest and lowest numbers of MQTL did not match.

  1. Line 266: Why the number of MQTL in your study is so much larger than the others? Your approach might have been too permissive. Therefore, you should describe better in methodology the threshold value for a CI to be considered significative.

R: The confidence interval (95%) to determine the MQTL was added in the methodology section. The higher number of MQTLs detected in this study compared to the other MQTL analyses performed on common bean is due to the higher number of individual QTL reported for yield related traits in different studies.

  1. Lines 270-284: The present work has no frame to explore the result in terms of drought tolerance. This is speculative and not proved by the analyze data. My advice is that this part of the discussion must be withdrawn (as well as in the Abstract). Please concentrate your discussion in Yield components which were the ones analyzed. There is no support in the candidate genes found that these MQTL are related to drought tolerance.

R: Thank you for this valuable comment. Previous studies have associated the Auxin response factors (ARFs) with drought tolerance. In consequence, the idea was maintained and it is speculative, and references were added to support this idea.

  1. The study lacks a final conclusion (paragraph). This section of the paper can be improved.

R: It was added, and a global conclusion is at point 5 according to the “instructions for the authors”.

Reviewer 2 Report

Dear Author, 

Thank you for your work. Yield-related traits are very important agronomy factors. The authors invested a large time to create populations and identify the QTLs.

 For your manuscript, I have a few comments.

 1. For table 2, it would be better to have an abbreviation list instead of a table.

2. For reference, authors check the reference pattern carefully, for instance, the year of publication needs bold on line 418.

3. For figure 2 looks a little bit fuzzy, which requests a higher resolution.

Thank you.

Author Response

Responses to reviewer 2

  1. For table 2, it would be better to have an abbreviation list instead of a table.

R: Table 2 was maintained; however, the traits abbreviations were explained in the materials and methods.

  1. For reference, authors check the reference pattern carefully, for instance, the year of publication needs bold on line 418.

R: It was done.

  1. For figure 2 looks a little bit fuzzy, which requests a higher resolution.

R: The resolution was improved, thank you very much for your remarks and suggestions.
